# Privacy-Preserving Event-Triggered Predefined Time Containment Control for Networked Agent Systems

1st Weihao Li†
*School of Aeronautics and Astronautics*
*University of Electronic Science and Technology of China*
Chengdu, China
liweihao@std.uestc.edu.cn

2nd Jiangfeng Yue†
*School of Aeronautics and Astronautics*
*University of Electronic Science and Technology of China*
Chengdu, China
yuejiangfeng_1@163.com

3rd Mengji Shi
*School of Aeronautics and Astronautics*
*University of Electronic Science and Technology of China*
Chengdu, China
maangat@126.com

4th Boxian Lin
*School of Aeronautics and Astronautics*
*University of Electronic Science and Technology of China*
Chengdu, China
linbx@uestc.edu.cn

5th Kaiyu Qin
*School of Aeronautics and Astronautics*
*University of Electronic Science and Technology of China*
Chengdu, China
kyqin@uestc.edu.cn

*Abstract*—**This paper addresses the privacy-preserving event-triggered predefined time containment control problem for networked agent systems. A novel containment control scheme is developed that integrates privacy protection with event-triggered mechanisms, optimizing network efficiency by minimizing unnecessary data transmission while ensuring robust containment within a specified time frame. The proposed control scheme ensures the confidentiality of agents' information through output masking, thereby maintaining both privacy and control accuracy. Furthermore, it provides a distinct advantage over traditional finite-time and fixed-time control methods by guaranteeing convergence to the desired state within a predefined time, regardless of initial conditions. Finally, some simulation results are given to verify the effectiveness of the proposed containment control scheme.**

*Index Terms*—**Containment Control; Privacy-preserving; Predefined Time; Event-triggered Control; Networked Agent Systems.**

## I. INTRODUCTION

Networked agent systems have garnered significant attention across various fields due to their broad range of applications, including robotics [1], autonomous vehicles [2], and distributed sensor networks [3]. The cooperative control of networked agent systems involves designing strategies that enable agents to work together effectively to achieve shared objectives. A prominent approach within cooperative control is containment control [4], [5], which aims to ensure that a group of agents (followers) remains within a specified region or adheres to a particular trajectory, while another group of agents (leaders) directs their behavior. Containment control is particularly crucial in scenarios requiring strict spatial or operational constraint adherence. For instance, in a formation flying scenario, containment control can ensure that a group of drones maintains a specific formation while another set of drones guides their collective movement [6].

Convergence speed is a critical performance metric in the containment control of networked agent systems. Current research explores several approaches to achieving convergence, including asymptotic convergence [7], finite-time convergence [8], and fixed-time convergence [9]. Asymptotic convergence guarantees that the system will eventually converge to the desired state over time, although the convergence rate may not be specified. Finite-time convergence ensures that the system reaches the desired state within a finite period, though the exact time depends on system parameters and states. Fixed-time convergence provides a guarantee of convergence within a predetermined time, irrespective of initial conditions, thereby offering more predictability in performance. However, the convergence time in both finite-time and fixed-time approaches is influenced by system parameters and states. To address this, researchers have developed predefined time control schemes that enable the specification of a desired convergence time [10], [11]. The primary advantages of predefined-time control

†:These authors contribute equally to this paper.

The authors are also with Aircraft Swarm Intelligent Sensing and Cooperative Control Key Laboratory of Sichuan Province, Chengdu, 611731, China and National Laboratory on Adaptive Optics, Chengdu, 610209, China. This work was supported by the National Natural Science Foundation of China (52102453), the Natural Science Foundation of Sichuan Province (2022NSFSC0037, 2024NSFSC0021), the Sichuan Science and Technology Programs (MZGC20230069, MZGC20240139), the Fundamental Research Funds for the Central Universities (ZYGX2024K028). (Corresponding author: *Mengji Shi*.)

include the ability to guarantee convergence within a specified time frame, thereby providing more predictable and controllable system behavior, and enhancing system performance by setting precise deadlines for achieving the desired state.

The existing literature [10]–[13] on predefined-time convergence in networked agent systems generally overlooks the issue of information privacy during transmission. However, privacy protection is of paramount importance in containment control, where safeguarding the confidentiality of agents' information is critical. Several methods for privacy protection have been proposed, including state decomposition [14], differential privacy [15], additive noise [16], and output masking [17]. Among these, output masking has received considerable attention due to its simplicity and ease of implementation. This method involves obscuring the output of agents to protect sensitive information while still allowing effective control. However, output masking relies on continuous information exchange, which can impose constraints on communication bandwidth. To address this limitation, it is necessary to develop privacy protection schemes under event-triggered mechanisms [18], [19], which can alleviate communication bandwidth constraints. In [19], the authors integrated both privacy preservation and event-triggered mechanisms into the consensus and containment control but overlooked predefined performance. Zhang et al. [20] incorporated prescribed-time theory and privacy preservation into consensus control but neglected bandwidth constraints. In conclusion, to the best of the author's knowledge, no existing solution simultaneously addresses the challenges of communication bandwidth, convergence time, and privacy protection in containment control, making this an area of significant research opportunity.

According to the above discussion, this paper focuses on the privacy-preserving event-triggered predefined time containment control problem of networked agent systems. The main contributions of this paper are summarized as follows:

(1) A novel event-triggered predefined-time containment control scheme is developed to optimize network efficiency while ensuring robust containment performance within a specified time frame. By employing event-triggered control, the scheme significantly reduces unnecessary data transmission, ensuring that agents communicate only when necessary. This approach effectively balances communication efficiency and system performance.

(2) The proposed control scheme guarantees convergence within a predefined time, offering a distinct advantage over finite-time and fixed-time methods. Unlike these traditional methods, where convergence time is often influenced by initial conditions and system parameters, the predefined time control ensures that the desired state is consistently reached within the predetermined time frame, thereby enhancing the predictability and reliability of the system.

(3) Furthermore, a privacy-preserving containment control scheme is designed to safeguard the confidentiality of agents' information by masking their outputs while maintaining accurate control. Compared to alternative privacy protection methods such as differential privacy or state decomposition,

this scheme provides a simpler and more efficient solution. It ensures both privacy and communication efficiency without compromising the overall system performance, making it particularly suitable for applications with stringent privacy and bandwidth requirements.

The remainder of the paper is listed below. Some preliminaries are formulated in Section II and Section III formulates the problem. Section IV designs a privacy-preserving containment control input. Numerical simulation examples are provided in Section V, and Section VI sums up the whole paper.

## II. PRELIMINARY AND PROBLEM FORMULATION

### A. Preliminaries

The communication structure among agents in this study is represented by a graph topology denoted as $\mathcal{G} = \langle \mathcal{V}, \mathcal{E}, \mathcal{A} \rangle$, where $\mathcal{V}$, $\mathcal{E}$, and $\mathcal{A}$ correspond to the set of nodes, the set of edges, and the adjacency matrix, respectively. The network consists of a total of $N = m + n$ agents, with $n$ being the number of follower agents and $m$ being the number of leader agents. The leader and follower agents are categorized into sets $\mathcal{V}_L = \{1, 2, \ldots, m\}$ and $\mathcal{V}_F = \{m+1, m+2, \ldots, m+n\}$, respectively. Consequently, the overall set of nodes is formed by the union of these two sets, $\mathcal{V} = \mathcal{V}_F \cup \mathcal{V}_L$. Following the definitions of the node sets, the adjacency matrix is represented as $\mathcal{A} = [a_{ij}] \in \mathcal{R}^{(n+m) \times (n+m)}$, where the element $a_{ij}$ is positive if there exists an edge from node $j$ to $i$ within the set $\mathcal{E}$, and zero otherwise. Assuming leaders do not have adjacent nodes, implying that leaders solely disseminate information to followers, the Laplacian matrix $\mathcal{L}$ for the network of agents is derived as $\mathcal{L} = \mathcal{D} - \mathcal{A}$. The degree matrix, denoted by $\mathcal{D}$, is a diagonal matrix with elements $d_i$ on the diagonal, where $d_i$ is the sum of the adjacency matrix elements in the $i$-th row, calculated as $d_i = \sum_{k=1}^{n+m} a_{ik}$.

Based on the aforementioned definitions, the Laplacian matrix is constructed as follows:

$$\mathcal{L} = \begin{bmatrix} \mathbf{0}_{m \times n} & \mathbf{0}_{m \times m} \\ \mathcal{L}_F & \mathcal{L}_L \end{bmatrix}, \tag{1}$$

where the sub-Laplacian matrix specific to the follower agents is denoted as $\mathcal{L}_F \in \mathcal{R}^{n \times n}$, and the sub-Laplacian matrix that captures the interactions between leader and follower agents is represented by $\mathcal{L}_L \in \mathcal{R}^{n \times m}$. The elements of $\mathcal{L}_F$, denoted as $[l_{ij}]$, are defined such that when indices match, $l_{ij}$ equals the sum of the adjacency matrix entries $a_{ip}$ for all $p$ in the set of nodes $\mathcal{V}$, and when indices differ, $l_{ij}$ is the negation of the corresponding adjacency entry $a_{ij}$. Mathematically, this is expressed as:

$$l_{ij} = \begin{cases} \sum_{p=1}^{m+n} a_{ip}, & \text{if } i = j, \\ -a_{ij}, & \text{otherwise.} \end{cases}$$

The subsequent assumption about the communication framework is established to guarantee the feasibility of containment control within the networked agent systems.

***Assumption 1:*** This paper assumes that each follower is associated with at least one leader, with whom there exists a directed path leading to the follower.

*Definition 1 ([21]):* Let $Z_n$ be the collection of all $n \times n$ square matrices with non-positive off-diagonal elements, denoted as $Z_n \subset \mathcal{R}^{n \times n}$. A matrix $Y$ is classified as a nonsingular M-matrix if it belongs to $Z_n$ and all its eigenvalues possess positive real parts.

*Lemma 1 ([4]):* Under Assumption 1, it is established that the matrix $\mathcal{L}_F$ qualifies as a nonsingular M-matrix. Furthermore, it holds that $-\mathcal{L}_F^{-1}\mathcal{L}_L \mathbf{1}_m = \mathbf{1}_n$, and every component of $-\mathcal{L}_F^{-1}\mathcal{L}_L$ is nonnegative.

*Definition 2 ([22]):* Let $\Lambda$ be a subset of $\mathcal{R}^n$. If for any $z_1, z_2 \in \Lambda$ and a scalar $0 < \gamma < 1$, the linear combination $(1-\gamma)z_1 + \gamma z_2$ also belongs to $\Lambda$, then $\Lambda$ is deemed a convex set. Given a vector $\chi$ with elements $\chi_i$, the convex hull of $\chi$, denoted as $\mathrm{Co}(\chi)$, is the set of all vectors that can be expressed as $\sum_{i=1}^{n} \gamma_i \chi_i$, where each $\gamma_i \geq 0$ and the sum $\sum_{i=1}^{n} \gamma_i = 1$.

### B. Time-varying transformation

The objective of privacy-preserving containment control is to guide the followers into the convex hull spanned by the leaders, without revealing the initial states of the participating agents. To address this, the paper integrates a dynamic, time-variant transformation into the traditional containment control paradigm. This transformation enables each agent to modify its state according to the evolving function before sharing information with its neighbors. The employed transformation function is both standardized and perpetually updating, characterized as

$$p : \mathcal{R}^+ \times \mathcal{R}^h \times \mathcal{R}^d \to \mathcal{R}^h$$
$$(t, x, m) \mapsto y(t) = \Lambda(t, x(t), m), \quad (2)$$

where $x = [x_1, \ldots, x_h]^T \in \mathcal{R}^h$ is the agent's true states, the hidden state output after the time-varying transformation is $y = [y_1, \ldots, y_h]^T \in \mathcal{R}^h$, both states have equal dimensions, the parameter set $m \in \mathcal{R}^d$ represents the key of time-varying transformation. The state output after the time-varying transformation is uniformly referred to as the hidden state in this paper. It is postulated that there exists a common system $\dot{x} = f(x)$, and the dynamics following the application of time-varying transformation can be expressed as $\dot{x} = f(y)$ and $y = \Lambda(t, x, m)$. If $|\Lambda(t, x, m) - x(t)|$ is approaching zero under the given key $m$, it is referred to as a finite time-varying transformation, and the following condition holds

$$\begin{cases} \lim_{t \to \Omega} \Lambda(t, x(t), m) = x(t), \\ \Lambda(t, x(t), m) = x(t), t \in [\Omega, \infty), \end{cases}$$

where $\Omega$ denotes a finite time constant indicates that the final hidden state converges to the real state over time. The range of $\Omega$ is primarily influenced by the values of each parameter in the key $m$.

### C. Containment control problem description

In this paper, we focus on a single-integrator networked agent system. The dynamics of the follower agents are characterized by the following equation:

$$\dot{x}_i(t) = u_i(t), \ i \in \mathcal{V}_F, \quad (3)$$

where $x_i(t)$ and $u_i(t)$ denote the position and control input of $i$th follower agent, respectively.

Additionally, the dynamics of the leader agents are governed by the following equation:

$$\dot{x}_i(t) = 0, \ i \in \mathcal{V}_L, \quad (4)$$

where $x_i(t)$ denotes the position of $i$th leader agent. The above dynamics mean that the leader agents' position is stationary.

*Definition 3:* Consider a single-integrator networked agent system comprising $m$ leader agents and $n$ follower agents, the implementation of predefined time containment control necessitates that the position states of the followers converge to the convex hull defined by the leaders within specified time $T$. Specifically, for any given initial condition, the convergence is characterized by the satisfaction of the following set of equations:

$$\lim_{t \to T} |x_i(t) - \sum_{k=1}^{m} \varepsilon_{ik} x_k(t)| = 0, \quad (5)$$

where $\varepsilon_{ik} \in \mathcal{R}, \varepsilon_{ik} \geq 0$ and $\sum_{k=1}^{m} \varepsilon_{ik} = 1, i \in \mathcal{V}_F, k \in \mathcal{V}_L$.

## III. MAIN RESULTS

To safeguard the confidentiality of agents' initial state information, we introduce mutually independent functions into the process of information exchange among agents. Furthermore, the aforementioned time-varying function can be implemented as

$$\begin{cases} \lim_{t \to T_i} \Lambda_i(t, x_i(t), m_i) = x_i(t), \\ \Lambda_i(t, x_i(t), m_i) = x_i(t), t \in [T_i, \infty). \end{cases} \quad (6)$$

According to the requirements of the finite-time varying function, the received information of follower agent $j$ from agent $i$ can be designed as

$$\begin{cases} \mathrm{R}_i^m(t) = \Lambda_i(t, x_i(t), m_i) \\ \Lambda_i(t, x_i(t), m_i) = x_i(t) + a_i t^2 + b_i t + c_i, & t \in [0, \Omega_i) \\ \Lambda_i(t, x_i(t), m_i) = x_i(t), & t \in [\Omega_i, \infty) \end{cases}$$

where $\Omega_i$ satisfies

$$\begin{cases} \Omega_i = \frac{-b_i - \sqrt{b_i^2 - 4a_i c_i}}{2a_i}, b_i \geq 0, c_i \geq 0, \text{if } a \in [0, \infty), \\ \Omega_i = \frac{-b_i + \sqrt{b_i^2 - 4a_i c_i}}{2a_i}, b_i < 0, c_i < 0, \text{if } a \in (-\infty, 0), \end{cases}$$

and $a_i, b_i, c_i \in \mathcal{R}$, each agent has its distinctive encode key, denoted as $m_i = \{a_i, b_i, c_i\}$, noting that individual encode keys remain undisclosed to other agents.

Building upon the previously devised time-varying function and the acquired hidden information from neighboring agents, the predefined time containment control input for the $i$th agent can be expressed as follows

$$\begin{cases} u_i(t) = -\left(\rho + \delta\frac{\dot{\mu}}{\mu}\right) \sum_{j \in \mathcal{V}_L \cup \mathcal{V}_F} a_{ij}\left(\mathrm{R}_i^m(t) - \mathrm{R}_j^m(t)\right), \\ \Lambda_i(t, x_i(t), m_i) = x_i(t) + a_i t^2 + b_i t + c_i, t \in [0, \Omega_i), \\ \Lambda_i(t, x_i(t), m_i) = x_i(t), t \in [\Omega_i, \infty), \end{cases} \quad (7)$$

where $\rho > 0$ represents the control gain, and $\mu$ denotes a time-varying scaling function, which takes the form of

$$\mu(t) = \begin{cases} \left(\frac{T}{T-t}\right)^h, & t \in [0, T), \\ 0, & t \in [T, \infty), \end{cases}$$

where the real number $h$ holds the condition $h > 2$.

Considering the practical challenges encountered in networked agent systems, which frequently involve communication limitations, the incorporation of an event-triggered mechanism can considerably reduce the utilization of communication resources. In this paper, we integrate the event-triggered mechanism into the aforementioned controller.

*Assumption 2:* When employing an event-triggered mechanism, it is presupposed that every agent has the capability to actively monitor its state information in real time. Furthermore, agents are designed to disseminate relevant state updates contingent upon the fulfillment of designed event-triggering condition.

To ensure synchronization among all agents, we establish a triggering sequence denoted as $\{t_1, t_2, \ldots, t_k\}$. This sequential arrangement guarantees that all agents update their controllers simultaneously at a unified triggering time. As a result, the control input (7) can be reformulated as

$$u_i(t) = -\left(\rho + \delta\frac{\dot{\mu}}{\mu}\right) \sum_{j \in \mathcal{V}_L \cup \mathcal{V}_F} a_{ij}\left(\mathrm{R}_i^m(t_k) - \mathrm{R}_j^m(t_k)\right). \quad (8)$$

For each agent, the state measurement error between triggering and true state is

$$e_i^m(t) = \mathrm{R}_i^m(t_k) - \mathrm{R}_i^m(t), t \in [t_k, t_{k+1}). \quad (9)$$

Substituting the state measurement error and the controller into the agent's dynamics, yields

$$\begin{aligned} \dot{x}_i(t) =& -\mathrm{K}_\rho \sum_{j \in \mathcal{V}_L \cup \mathcal{V}_F} a_{ij}\left(\mathrm{R}_i^m(t_k) - \mathrm{R}_j^m(t_k)\right) \\ =& -\mathrm{K}_\rho \sum_{j \in \mathcal{V}_L \cup \mathcal{V}_F} a_{ij}\left(e_i^m(t) + \mathrm{R}_i^m(t) - \left(e_j^m(t) + \mathrm{R}_j^m(t)\right)\right) \\ =& -\mathrm{K}_\rho \sum_{j \in \mathcal{V}_L \cup \mathcal{V}_F} a_{ij}\left(e_i^m(t) - e_j^m(t)\right) \\ & -\mathrm{K}_\rho \sum_{j \in \mathcal{V}_L \cup \mathcal{V}_F} a_{ij}\left(\mathrm{R}_i^m(t) - \mathrm{R}_j^m(t)\right), \end{aligned}$$

where $\mathrm{K}_\rho = \rho + \delta\frac{\dot{\mu}}{\mu}$, and its corresponding compact form can be represented as

$$\begin{aligned} \dot{x}(t) =& -\mathrm{K}_\rho \mathcal{L}\mathrm{R}^m(t) - \mathrm{K}_\rho \mathcal{L}e^m(t) \\ =& -\mathrm{K}_\rho \left(\mathcal{L}_F(\mathrm{R}_F^m(t) + e_F^m(t)) + \mathcal{L}_L(\mathrm{R}_L^m(t) + e_L^m(t))\right). \end{aligned}$$

where $x(t) = \mathbf{col}_i^{n+m}[x_i(t)]$, $\mathrm{R}_F^m(t) = \mathbf{col}_i^n[\mathrm{R}_{Fi}^m(t)]$, $\mathrm{R}_L^m(t) = \mathbf{col}_i^m[\mathrm{R}_{Li}^m(t)]$, $e_L^m(t) = \mathbf{col}_i^m[e_{Li}^m(t)]$ and $e_F^m(t) = \mathbf{col}_i^n[e_{Fi}^m(t)]$. Besides, let $A = \mathbf{col}_i^{n+m}[a_i]$, $B = \mathbf{col}_i^{n+m}[b_i]$ and $C = \mathbf{col}_i^{n+m}[c_i]$.

Accordingly, the whole closed-loop error system is

$$\begin{cases} \dot{x}(t) = -\mathrm{K}_\rho \mathcal{L}\mathrm{R}^m(t) - \mathrm{K}_\rho \mathcal{L}e^m(t) \\ \mathrm{R}^m(t) = x(t) + m(t) \end{cases} \quad (10)$$

where

$$m(t) = \begin{cases} At^2 + Bt + C, & t \in [0, T^1) \\ A_{m_1}t^2 + B_{m_1}t + C, & t \in [T^1, T^2) \\ \vdots \\ A_{m_1 \ldots m_{N-1}}t^2 + B_{m_1 \ldots m_{N-1}}t + C, & t \in [T^{N-1}, T^N) \\ 0, & t \in [T^N, \infty) \end{cases}$$

To address the predefined time privacy-preserving containment control under the event-triggered mechanism, we design the event-triggering condition (ETC) for the networked agent systems as

$$t_{k+1} = \inf\left\{t > t_k : \|e^m(t)\| \geq (1-\varepsilon)\frac{\mathrm{K}_\rho^\lambda}{\mathrm{K}_\rho}\frac{\|\varpi(t)\|}{\|\mathcal{L}\|}\right\}. \quad (11)$$

where $\mathrm{K}_\rho = \rho + \delta\frac{\dot{\mu}}{\mu}$ and $\mathrm{K}_\rho^\lambda = \rho\lambda_2(\mathcal{L}_F) + \delta\frac{\dot{\mu}}{\mu}$, $\varepsilon \in (0,1)$ and $\lambda_2(\mathcal{L}_F)$ is the second smallest eigenvalue of the Laplacian matrix $\mathcal{L}_F$. Upon the occurrence of a triggering event, all agents discard their previous state and proceed to sample their current state to update their controller. Subsequently, they transmit the newly sampled state to their neighboring agents. Throughout the inter-event period, their control inputs remain constant until the next triggering instance, which forcibly violates the event-triggering condition.

*Theorem 1:* Under the event-triggering condition (11) and control input (8), the predefined time privacy-preserving containment control for networked agent system with graph $\mathcal{G}$ can be achieved. While the parameter in ETC satisfies $\varepsilon \in (0,1)$.

**Proof:** The proof of Theorem 1 includes convergence analysis and privacy analysis, respectively.

**(I) Convergence analysis:** The vector $x(t)$ can be divided into sub-vector $x_F(t)$ and $x_L(t)$. Based on Definition 3, we define the containment error as $\varpi(t) = x_F(t) - \left(-\mathcal{L}_F^{-1}\mathcal{L}_L x_L(t)\right)$, and Lyapunov function is adopted as

$$V(t) = \varpi(t)^T\varpi(t). \quad (12)$$

Note that the leader agents' dynamics model (4), it yields

$$\dot{\varpi}(t) = \dot{x}_F(t) - \left(-\mathcal{L}_F^{-1}\mathcal{L}_L\dot{x}_L(t)\right) = \dot{x}_F(t).$$

Taking the derivative of the Lyapunov function $V(t)$, one obtains the following expression

$$\begin{aligned} \dot{V}(t) =& \varpi(t)^T\dot{\varpi}(t) = \varpi(t)^T\dot{x}_F(t) \\ =& \varpi(t)^T\left(-\mathrm{K}_\rho\left(\mathcal{L}_F(\mathrm{R}_F^m(t) + e_F^m(t)) + \mathcal{L}_L(\mathrm{R}_L^m(t) + e_L^m(t))\right)\right) \\ =& -\rho\varpi(t)^T\left(\mathcal{L}_F(\mathrm{R}_F^m(t) + e_F^m(t)) + \mathcal{L}_L(\mathrm{R}_L^m(t) + e_L^m(t))\right) \\ & -\delta\frac{\dot{\mu}}{\mu}\varpi(t)^T\left(\mathcal{L}_F(\mathrm{R}_F^m(t) + e_F^m(t)) + \mathcal{L}_L(\mathrm{R}_L^m(t) + e_L^m(t))\right). \end{aligned}$$

To satisfy the privacy-preserving requirement of designing a time-varying transformation function, it is essential to ensure that $T^N$, the moment at which the final time-varying function converges to its corresponding true state, is less than $T$, for all $t \in [0, T)$. Notably, the value of $m(t)$ decreases monotonically as $t$ increases in the interval $t \in [0, T^N)$, and it attains zero if $t \in [T^N, T)$. The result further derives the condition

$\lim_{t \to T_N} R_F^m(t) = x_F(t), \lim_{t \to T_N} R_L^m(t) = x_L(t)$. Based on Lemma 1 in [11], it follows that

$$\mathcal{L}_F(x_F(t) + e_F^m(t)) + \mathcal{L}_L(x_L(t) + e_L^m(t))$$
$$= \mathcal{L}_F \left( (x_F(t) + e_F^m(t)) + \mathcal{L}_F^{-1}\mathcal{L}_L(x_L(t) + e_L^m(t)) \right)$$
$$= \mathcal{L}_F \left( x_F(t) + \mathcal{L}_F^{-1}\mathcal{L}_L x_L(t) \right) + \mathcal{L}_F e_F^m(t) + \mathcal{L}_L e_L^m(t)$$
$$= \mathcal{L}_F \varpi(t) + \mathcal{L} e^m(t).$$

It is noted that $\mathcal{L}_F \in \mathcal{R}^{n \times n}$ denotes the sub-Laplacian matrix among follower agents, we can obtain $\varpi(t)^T \mathcal{L}_F \varpi(t) \le \lambda_2(\mathcal{L}_F)\varpi(t)^T\varpi(t)$, and it derives

$$\dot{V}(t) \le -K_\rho^\lambda V(t) - K_\rho\varpi(t)^T\left(\mathcal{L}_F e_F^m(t) + \mathcal{L}_L e_L^m(t)\right)$$
$$= -\varepsilon K_\rho^\lambda V(t) - (1-\varepsilon)K_\rho^\lambda V(t) - K_\rho\varpi(t)^T\mathcal{L}e^m(t)$$
$$\le -\varepsilon K_\rho^\lambda V(t) - (1-\varepsilon)K_\rho^\lambda\|\varpi\|^2 + K_\rho\|\varpi\|\|\mathcal{L}e^m\|.$$

Considering the designed event-triggering condition (11) and the condition $\varepsilon \in (0,1)$, it concludes

$$K_\rho\|\mathcal{L}e^m(t)\| \le (1-\varepsilon)K_\rho^\lambda\|\varpi(t)\|.$$

Accordingly, since $\delta \ge 1$, it yields

$$\dot{V}(t) \le -\left(\rho\lambda_2(\mathcal{L}_F) + \frac{\dot{\mu}}{\mu}\right)\varpi(t)^T\varpi(t) = \rho\lambda_2(\mathcal{L}_F)V - \frac{\dot{\mu}}{\mu}V.$$

According to the Lemma 1 in [11], one has

$$V(t) \le \mu(t)^{-2}\exp^{-\rho\lambda_2(\mathcal{L}_F)(t-T^N)}V\left(T^N\right). \qquad (13)$$

And then $\|\varpi(t)\| \le \mu(t)^{-1}\exp^{-\rho\lambda_2(\mathcal{L}_F)(t-T^N)}\|\varpi(T^N)\|$. Note that $\lim_{t \to T^-}\mu(t)^{-1} = 0$, it yields $\lim_{t \to T^-}\|\varpi(t)\| = 0$. That is, when $t \to T^-$, the condition $x_F(t) - (-\mathcal{L}_F^{-1}\mathcal{L}_L x_L(t)) = 0$ holds. Based on the equation (46) of [19] and Definition (2)-(3), $-\mathcal{L}_F^{-1}\mathcal{L}_L x_L(t)$ is the convex hull signal formed by the leaders, when $\varpi(t) = 0$, it implies that all followers converge within the convex hull formed by the leaders. Therefore, the containment control of the networked agent system is achieved within the predefined time $T$. Since the finite time-varying transformation is only applied to the interval $[0, T)$, the problem of predefined-time containment can be transformed into the general case discussed in [11] for $t \in [T, \infty)$. For further information, interested readers can refer to Theorem 1 in [11], which provides detailed proof.

**(II) Privacy analysis:** Consider a scene where the dynamics $f(\cdot)$ of all agents are widely known and each agent has access to the hidden output states $R_i^m(t)$ of its neighboring agents. While the true states $x_i(t)$ and the encode keys $\{a_i, b_i, c_i\}$ are regarded as private information exclusive to each agent. For an honest-but-curious agent, the information accessible includes the unsigned graph $\mathcal{G}$, the state of the honest-but-curious agents and the set of neighboring agents, and the hidden state of both the honest-but-curious agents and their neighbors. Following the application of a finite time-varying transformation to conceal agent $i$'s initial state, the resulting hidden output $R_i^m(t)$ bears no resemblance to the true initial value $x_i(0)$. As a result, any information set acquired by an honest but curious agent proves futile in determining agent $i$'s true initial state. Additionally, the agent cannot reconstruct

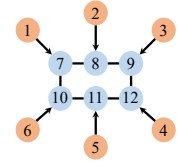

Fig. 1. The communication topology among twelve agents.

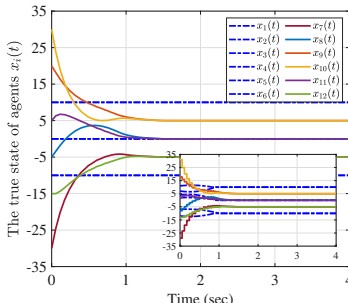

Fig. 2. The true and masked states of all agents.

their true initial state by employing the findings presented in [23]. Importantly, even external eavesdroppers are unable to obtain the true initial state, as evidenced by the process mentioned above. Thus, it becomes apparent that the integrity of the initial state remains elusive to all parties involved, substantiating the claim of its unattainability by external eavesdroppers.

## IV. SIMULATION

In this section, several numerical simulations are conducted to verify the effectiveness of the theoretical analysis. The simulation consists of the networked agent systems comprising 12 agents, which include six followers and six leaders. Fig. 1 displays the communication topology among agents. The numerical simulations are performed in the 2-D space. The initial position states of all agents are set as $x^1(0) = [-10, 0, 10, 10, 0, -10, -30, -5, 20, 30, 5, -15]^T$ and $x^2(0) = [5, 5, 5, -5, -5, -5, 5, 20, 25, -10, -15, -20]^T$. And the parameter $\varepsilon$ is equal to 0.5, the predefined time is $T = 1.5s$. The encode keys are selected as

$$A = [-5, -9, -5, 8, -3, 6, -4, 5, 6, -4, 5, -3]^T,$$
$$B = [2, 4, 3, -4, 1, -3, 2, -1, -3, 2, -1, 1]^T,$$
$$C = [3, 4, 1, -3, 2, -2, 1, -3, -2, 1, -3, 2]^T.$$

The simulation results are depicted in Fig 2-4. The trajectory of agents in the $x^1$ direction is illustrated in Fig 2, with the subfigure highlighting the masked trajectories of all agents. This indicates that the proposed method effectively preserves the privacy of the agents' initial states and achieves predefined time convergence within $1.5s$. Fig 3 demonstrates the fulfillment of the event-triggering conditions, when the designed boundary threshold is exceeded, the agents' states are sampled and updated. Fig 4 shows that all followers successfully move from their initial positions into the convex

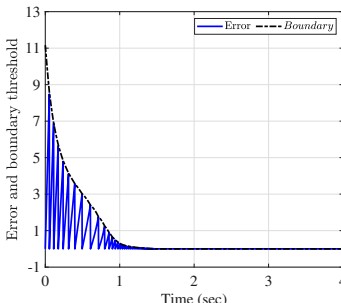

Fig. 3. The trajectory of the state measurement error and boundary threshold.

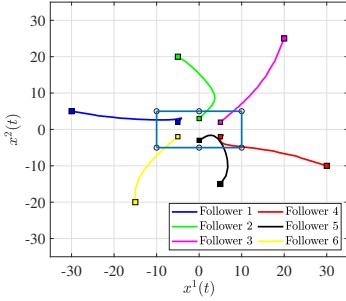

Fig. 4. The trajectory of all agents in the 2-D plane under designed containment control input. (Square markers represent the followers, and circular markers represent the leaders. Leaders form a rectangular convex hull.)

hull formed by the fixed leaders, achieving privacy-preserving event-triggered predefined time containment control for the networked agent system.

## V. CONCULSION

This paper has addressed the privacy-preserving event-triggered predefined-time containment control problem for networked agent systems. A novel containment control scheme has been developed, effectively integrating privacy protection with event-triggered mechanisms. This integration has optimized network efficiency by minimizing unnecessary data transmission while ensuring robust containment within a specified time frame. The proposed control scheme has successfully ensured the confidentiality of agents' information through output masking, thereby maintaining both privacy and control accuracy. The effectiveness of the proposed scheme has been verified through simulation results. It is important to note that this study has focused on static leaders, and future research will extend the investigation to address containment control problems under dynamic leaders.

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
