# OpenReview forum: "Privacy-Preserving Event-Triggered Predefined Time Containment Control for Networked Agent Systems"
_IEEE.org/ICIST/2024/Conference — IEEE ICIST 2024 Conference Submission_

### Official Review · Reviewer_CpBr · 2024-08-21
**Accept**

**Rating:** 7
**Confidence:** 5

**Review:**

This manuscript addresses the privacy-preserving event-triggered predefined time containment control problem in networked agent systems. It proposes a control scheme that combines privacy protection with event-triggered mechanisms, maintaining both the confidentiality of the agents and control accuracy. The effectiveness of the proposed algorithm is validated through simulation examples. The following suggestions should be considered to further improve the quality of the paper: 1. The abstract should be more concise to enhance readability. 2. The simulation section could include a comparison with predefined time control methods to validate the effectiveness of the proposed approach.

---

### Official Review · Reviewer_3g8X · 2024-08-25
**new insight and opinion into the development of privacy-preserving containment control for networked agent systems**

**Rating:** 8
**Confidence:** 4

**Review:**

In the manuscript titled"Privacy-Preserving Event-Triggered Predefined Time Containment Control for Networked Agent Systems"presents a privacy-preserving event-triggered containment control scheme for networked agent systems that minimizes data transmission, ensures confidentiality through output masking, and guarantees convergence to the desired state within a predefined time, with its effectiveness demonstrated through simulations.This work provides new insight and opinion into the development of privacy-preserving containment control for networked agent systems. The manuscript is well-organized and clearly stated. I would suggest accepting it after the following major/minor concerns are addressed:Further emphasis on research motivation;Increase future research directions.

---

### Official Review · Reviewer_AZqX · 2024-08-25
**This paper is about the predefined time control problem triggered by privacy-preserving events in network agent systems. The solution is to develop a new containment control scheme. The advantage is that it offers a distinct advantage over traditional methods, guaranteeing convergence to the desired state within a predefined time, regardless of the initial conditions. However.1. Highlight the innovation of the new event triggering mechanism used in this paper compared to the traditional event triggering. 2. Add future work and explain the direction of future improvement.**

**Rating:** 8
**Confidence:** 4

**Review:**

This paper is about the predefined time control problem triggered by privacy-preserving events in network agent systems. The solution is to develop a new containment control scheme that combines privacy protection with an event-triggering mechanism to optimize network efficiency by minimizing unnecessary data transfers while ensuring strong containment within a specified time frame. The proposed control scheme ensures the confidentiality of proxy information through output masking, so as to ensure privacy and control accuracy. The advantage is that it provides a distinct advantage over traditional finite and fixed-time control methods, guaranteeing convergence to the desired state within a predefined time, regardless of the initial conditions.
However.1. Highlight the innovation of the new event triggering mechanism used in this paper compared to the traditional event triggering.
2. Add future work and explain the direction of future improvement.

---

### Decision · Program_Chairs · 2024-09-08

Accept (Oral)